# Anti-Tumor Efficacy of PD-L1 Targeted Alpha-Particle Therapy in a Human Melanoma Xenograft Model

**DOI:** 10.3390/cancers13061256

**Published:** 2021-03-12

**Authors:** Marisa Capitao, Justine Perrin, Sylvain Simon, Sébastien Gouard, Nicolas Chouin, Frank Bruchertseifer, Alfred Morgenstern, Latifa Rbah-Vidal, Michel Chérel, Emmanuel Scotet, Nathalie Labarrière, Yannick Guilloux, Joëlle Gaschet

**Affiliations:** 1Université de Nantes, CNRS, Inserm, CRCINA, F-44000 Nantes, France; marisa.f.c@orange.fr (M.C.); justine.perrin@univ-nantes.fr (J.P.); ssylvain@fredhutch.org (S.S.); sebastien.gouard@univ-nantes.fr (S.G.); Latifa.Rbah-Vidal@univ-nantes.fr (L.R.-V.); Emmanuel.Scotet@univ-nantes.fr (E.S.); Nathalie.Labarriere@univ-nantes.fr (N.L.); yannick.guilloux@univ-nantes.fr (Y.G.); 2Université de Nantes, CNRS, Inserm, Oniris, CRCINA, F-44000 Nantes, France; nicolas.chouin@oniris-nantes.fr; 3European Commission, Joint Research Centre, Directorate for Nuclear Safety and Security, G-76344 Karlsruhe, Germany; frank.BRUCHERTSEIFER@ec.europa.eu (F.B.); Alfred.MORGENSTERN@ec.europa.eu (A.M.); 4Université de Nantes, CNRS, Inserm, ICO Gauducheau, GIP Arronax, CRCINA, F-44000 Nantes, France; Michel.Cherel@univ-nantes.fr

**Keywords:** TAT, PD-L1, melanoma, bismuth-213, theranostics

## Abstract

**Simple Summary:**

In recent years, the development of immune checkpoint inhibitors, such as anti-PD‑1 and anti-PD-L1, proved to prolong melanoma patient survival and are now used in routine clinical practice. PD-L1 also represents a potent biomarker for in vivo molecular imaging using radiolabeled anti-PD-L1 mAbs and positron emission tomography and is currently in development to select patients and assess response to treatment. The aim of our study was to investigate in a preclinical model of human melanoma if PD-L1 could also be a good target for treatment using targeted alpha-particle therapy. Our results show that targeting PD-L1 with bismuth-213, an alpha particle emitter, was associated with efficient anti-tumor response, significant tumor growth delay, and improved survival. This demonstrates that anti-PD-L1 antibodies could be used as theranostics in molecular imaging but also in targeted alpha-particle therapy to treat the tumor and its stroma.

**Abstract:**

PD-L1 (programmed death-ligand 1, B7-H1, CD274), the ligand for PD-1 inhibitory receptor, is expressed on various tumors, and its expression is correlated with a poor prognosis in melanoma. Anti-PD-L1 mAbs have been developed along with anti-CTLA-4 and anti-PD-1 antibodies for immune checkpoint inhibitor (ICI) therapy, and anti-PD-1 mAbs are now used as first line treatment in melanoma. However, many patients do not respond to ICI therapies, and therefore new treatment alternatives should be developed. Because of its expression on the tumor cells and on immunosuppressive cells within the tumor microenvironment, PD-L1 represents an interesting target for targeted alpha-particle therapy (TAT). We developed a TAT approach in a human melanoma xenograft model that stably expresses PD-L1 using a ^213^Bi-anti-human-PD-L1 mAb. Unlike treatment with unlabeled anti-human-PD-L1 mAb, TAT targeting PD-L1 significantly delayed melanoma tumor growth and improved animal survival. A slight decrease in platelets was observed, but no toxicity on red blood cells, bone marrow, liver or kidney was induced. Anti-tumor efficacy was associated with specific tumor targeting since no therapeutic effect was observed in animals bearing PD-L1 negative melanoma tumors. This study demonstrates that anti-PD-L1 antibodies may be used efficiently for TAT treatment in melanoma.

## 1. Introduction

PD-L1 (programmed death-ligand 1, B7-H1, CD274) is the primary ligand for PD-1 inhibitory receptor [1]. PD-L1 is constitutively expressed among various immune cells such as T-cells, B-cells, macrophages, and dendritic cells (DC) and upregulated upon activation [2,3]. Its function is not limited to inhibiting effector T-cells, as PD-L1 can induce Tregs in the periphery (iTregs) and sustain their suppressive functions [4]. PD-L1 is also expressed by various tumor cells, such as melanoma, non-small-cell lung carcinoma (NSCLC), triple-negative breast cancer (TNBC), as well as within the tumor microenvironment, on tumor-associated macrophages (TAM), and myeloid-derived suppressor cells (MDSC) **[5,6,7,8,9]**. In this context, PD-L1 expression results from pro-inflammatory stimuli and is subjected to a complex regulation [10,11]. In melanoma, it has been shown that IFN**γ** produced by tumor infiltrating T-lymphocytes (TILs) is the main factor inducing expression of PD-L1 [12,13]. Indeed, in biopsies of melanoma patients, colocalization of PD-L1 expressing tumor cells, CD8+ TILs and IFN**γ,** has been observed, and heterogenous expression of PD‑L1 was related to the T-cell infiltrate [14]. In addition, hypoxia, a frequent feature of solid tumors, and hypoxia inducible factor, HIF-1α, may also contribute to PD-L1 regulation in tumors and their microenvironment [8,15]. PD-L1 expression is considered to be one of the major tumor escape mechanisms and has been correlated with a more aggressive phenotype [16,17,18]. Several anti-PD-L1 mAbs—atezolizumab, avelumab, and durvalumab—have been developed as immune checkpoint inhibitor (ICI) therapy molecules and, based on their promising results in patients, have been recently approved for clinical use by the FDA and EMA [19,20,21].

Accordingly, with early clinical trials of ICI therapy using anti-PD-1 mAb, a review by Gandini et al. analyzed the clinical results of 4230 melanoma patients and reported an objective response rate of 48% for patients whose tumors expressed PD-L1 vs. 15% for those whose tumors were negative for PD-L1, supporting PD-L1 as a potent biomarker to select forefront the patients who can benefit from ICI therapy and to evaluate response to treatment [22,23,24,25]. However, numerous therapeutic responses have also been observed despite undetectable PD-L1 expression in the tumor [26,27]. So far, PD-L1 is determined by immunohistochemistry (IHC), which has several limitations related to the heterogeneous expression of the molecule [28,29]. IHC analysis has also been done using different mAbs, specific for various epitopes [14]. As a consequence, the relationship between PD-L1 minimum expression rate and therapeutic response is variable across studies, ranging from 1% to 50%, which actually makes PD-L1 not a clear predictive biomarker. To overcome IHC limits, numerous clinical trials are currently awaited or ongoing to assess reliability and sensitivity of in vivo molecular imaging using anti-PD-L1 mAbs radiolabeled with positron emitting radionuclides, that is, immuno-PET (positron emission tomography) imaging [30]. In a preclinical study evaluating PD-L1 as target for molecular imaging in an immunocompetent breast cancer model, Josefsson et al. provided biodistribution and dosimetry data supporting the feasibility of using a radiolabeled anti-PD-L1 mAb not only for molecular imaging but also for targeted radionuclide therapy (TRT) [31]. The ideal radionuclide to use for such TRT application would associate high toxicity to kill the tumor cells and the immunosuppressive cells within the tumor microenvironment, a short range of action to preserve surrounding healthy tissues, and the potency of stimulating immunity to provide an immune modulation.

Targeted alpha-particle therapy (TAT) is a TRT modality based on the use of alpha-particle emitters delivered specifically to the tumor by the mean of a vector, usually a mAb or a peptide. The alpha decay energy is comprised between 5 to 9 MeV along a short linear path in the tissues ranging from 50 to 100 μm; their linear energy transfer (LET) is therefore very high (50 to 230 keV/μm) and provides a high cytotoxic potential. Moreover, the radiobiological effects of this type of radionuclides are largely independent of dose rate, oxygenation, or cell proliferation [32]. Therefore, alpha-particle emitters are considered as interesting anti-tumor agents for micrometastases, residual tumors, and hematological cancers [33]. TAT in a preclinical immunocompetent melanoma model using anti-melanin mAb radiolabeled with bismuth-213 (^213^Bi), an alpha-particle emitter, has been shown to significantly delay tumor growth [34]. In the clinic, promising results have also been obtained in melanoma patients using ^213^Bi-anti-MCSP (melanoma-associated chondroitin sulfate proteoglycan) mAb. In a first clinic trial involving 16 patients, the radiolabeled mAb was delivered locally in melanoma lesions. In this context, TAT induced massive cell death with no toxicity [35]. In a second trial, the same radiopharmaceutical was delivered systemically in 38 patients developing metastatic melanoma. No major toxicity was observed, and the maximum tolerated dose (MTD) was not reached. An objective partial response was observed in 10% of the patients, and 15% survived more than 3 years [36,37].

Considering that PD-L1 expression is correlated with a poor prognostic in melanoma [38,39], that despite very promising clinical results, at best, 40% of metastatic melanoma patients objectively respond to ICI therapy targeting either PD-1 or PD-L1 [40,41], and that PD-L1 expression in the tumor and its stroma makes it a most interesting target for TAT, the aim of this study was to develop and investigate TAT using ^213^Bi-anti-PD-L1 mAb in a preclinical model of human melanoma.

## 2. Materials and Methods

### 2.1. Cells and Reagents

The M113^WT^ and M113^PD-L1+^ human melanoma cell lines were kindly provided by Dr. Nathalie Labarrière (CRCINA, Nantes, France). The cell lines were cultured in RPMI-1640 medium (Gibco, ThermoFisher Scientific, Waltham, MA, USA), supplemented with 2 mM L-glutamine (Gibco, ThermoFisher Scientific, Waltham, MA, USA), 100 U/mL penicillin (Gibco, ThermoFisher Scientific, Waltham, MA, USA), 100 ug/mL streptomycin (Gibco, ThermoFisher Scientific, Waltham, MA, USA), 10% heat-inactivated fetal bovin serum (Biosera Europe, Nuaille, France). M113^PD-L1+^ transfected cells were also supplemented by addition of 0.8 μg/mL of G418. Cells were incubated at 37 °C, 5% CO_2_, in a humidified-saturated incubator.

The GoInVivo™ purified anti-human PD-L1 mouse mAb (anti-hPD-L1 mAb), PE‑conjugated anti-hPD-L1 mAb (clone 29E.2A3), GoInVivo™ purified mouse IgG2bκ isotype control, and PE-conjugated mouse IgG2bκ isotype control (clone MPC-11) were purchased from Biolegend (San Diego, CA, USA). PE-conjugated goat anti-mouse IgG, F(ab’)_2_ was purchased from Jackson Immunoresearch (West Grove, PA, US). Flow cytometry experiments were performed using a FACS Canto II flow cytometer (BD Biosciences, San Jose, CA, USA), and the events were analyzed using the FlowJo software (Treestar, Meerhout, Belgium).

Before bismuth-213 radiolabeling, the anti-hPD-L1 mAb and the mouse IgG2bκ isotype control were modified using 2-(4-isothiocyanato-benzyl)-cyclohexyl-diethylenetriaminepenta-acetic acid (SCN-CHX-A’’-DTPA; Macrocyclics, Plano, TX, USA) with 20 equivalents of CHX-A’’-DTPA in carbonate buffer (0.05 M, pH 8.7). After 12h incubation at 25 °C, modified mAbs were purified by high-performance liquid chromatography on a Sephadex G200 gel-filtration column (Amersham Biosciences, Little Chalfont, UK). The mean chelate number per antibody, assessed as previously described [42], was 2. For bismuth-213 radiolabeling, 100 μg of each immunoconjugate was incubated with bismuth-213 eluted from an actinium‑225/bismuth-213 generator (Institute for Transuranium Elements, Karlsruhe, Germany) for 10 min at 37 °C in 0.8 M ammonium acetate (pH 5.3), 1.5% ascorbic acid. The resulting ^213^Bi-labelled immunoconjugates were separated from unbound bismuth-213 by size-exclusion chromatography using a PD-10 column (GE Healthcare, Chicago, IL, USA). Radiochemical purity was 98.7 ± 1.1%, as determined by instant thin-layer chromatography silica gel (ITLC-SG) using 0.1 M sodium citrate solution (pH 5.3) as mobile phase.

### 2.2. Mouse Xenograft Model

Female NSG (NOD.Cg-Prkdcscid Il2rgtm1Wjl/SzJ) mice were purchased from Charles River laboratory, housed and bred at the UTE animal facility (SFR François Bonamy, IRS‑UN, University of Nantes, license number: B-44-278) under specific pathogen free conditions. Subcutaneous xenograft tumors were established by injection of 1 × 10^6^ M113^PD-L1+^ or M113^WT^ human melanoma cells in 100 µL PBS (DPBS, ThermoFisher Scientific, Waltham, MA, USA), into the flank of 8–9-week-old NSG mice. Seven days later, when tumor volumes reached around 80 mm^3^, mice were randomly allocated into the different experimental groups. For each experiment, frozen M113^PD-L1+^ or M113^WT^ cells were thawed and grown in culture for 10 days before graft. Meanwhile, the absence of mycoplasma contamination was checked using a HEK-Blue Detection Kit (Invivogen, Toulouse, France), and PD-L1 expression was confirmed by flow cytometry analysis.

### 2.3. Histology and Immunohistochemistry Staining

Tumors were collected, formalin-fixed, and paraffin-embedded. Hematoxylin and eosin or immunohistochemistry staining were performed on 3µm paraffin sections. Expression of PD-L1 was analyzed using rabbit anti-human-PD-L1 (E1L3N^®^, Cell Signaling, Danvers, MA, USA) or rabbit isotype control (Cell Signaling, Danvers, MA, USA) primary antibodies and then using HRP-conjugated secondary antibody. Proliferation was analyzed using mouse anti-Ki67 (Clone MIB-1, Dako, Les Ulis, France) or mouse isotype control (BD Biosciences, San Jose, CA, USA) primary mAbs and HRP-conjugated secondary antibody. Revelation was done using DAB substrate solution and the sections were counterstained with hematoxylin. Acquisitions were performed using a slide scanner (Nanozoomer, Hamamatsu, Massy, France). For quantification, ndpi files were imported in QuPath version 0.2.3, a free software for the analysis of IHC images [43]. Simple tissue detection was performed on each slide, and extra-tumoral vessels were manually removed from the selection. Cell detection was performed by the detection of the nucleus, based on optical density sum. Ki67 positive cell was defined with a nucleus DAB (3,3′-Diaminobenzidine) mean staining equal or above a threshold of 0.3. Control isotypic stained slides showed a maximum of 1.15% positive cells with this threshold.

### 2.4. Immuno-PET Imaging

Copper-64 was obtained from the Arronax cyclotron (GIP Arronax, Saint-Herblain, France) using the reaction ^64^Ni(p,n)^64^Cu and was delivered as ^64^CuCl_2_ in 0.1N HCl. Before copper-64 radiolabeling, the anti-hPD-L1 mAb was modified using the copper chelating agent p-SCN-Bn-DOTA (Macrocyclics, Plano, TX, USA), purified on gel filtration column, and then radiolabeled for 30 min at 42 °C in 0.1M sodium acetate. The radiolabeling yield and specific activity of the bioconjugate were 100% and 482 MBq/mg, respectively. Each mouse was intravenously injected with 20 μg of ^64^Cu-labelled anti-hPD-L1 mAb. Total activity injected for each mouse was 10 MBq. Immuno-PET were realized on a multi-modality preclinical imaging system (Inveon™, Siemens Healthcare, Erlangen, Germany) at 24h and 48h after ^64^Cu-labelled immunoconjugate injection. The reconstructed PET images were analyzed using Inveon Research Workplace (Siemens Healthcare, Erlangen, Germany).

### 2.5. Dose-Escalation Study

Naïve 10–11-week-old NSG mice received an i.v. injection in the vein tail of 125, 165, 210, 335, and 395 kBq/g ^213^Bi-PD-L1 human mAb (*n* = 3 per group) and monitored for 100 days. Experiment was approved by the local veterinary committee (APAFIS #7915) and carried out in accordance with relevant guidelines and regulations. Animals were sacrificed in case of marked distress signs or/and a weight loss greater than 20% of initial body weight.

### 2.6. Therapy Studies

TAT studies were performed on mice bearing M113^PD-L1+^ or M113^WT^ tumors. Seven days after tumor graft, animals were treated by i.v. injection in the tail vein of a single dose of either 125 kBq/g (*n* = 17) or 165 kBq/g (*n* = 10) ^213^Bi-anti-hPD-L1 mAb. TAT control groups received either 125 kBq/g (*n* = 10) or 165 kBq/g (*n* = 10) ^213^Bi-mouse IgG2bκ isotype control. Animals treated with 125 kBq/g ^213^Bi-anti-hPD-L1 mAb or ^213^Bi-mouse IgG2bκ isotype control received around 6 μg of radiolabeled mAb, and those treated with 165 kBq/g ^213^Bi‑anti-hPD-L1 mAb or ^213^Bi-mouse IgG2bκ isotype control received around 10 μg of radiolabeled mAb. Finally, PBS control group received only an injection of 100 μL PBS (*n* = 20).

Experiments were approved by the local veterinary committee (APAFIS #7823) and carried out in accordance with relevant guidelines and regulations. Animals were monitored two to three times a week. Tumor burden was measured using a caliper, and the volume was calculated based the following formula: volume = (LxW^2^)/2, where L was length and W was width. Mice were sacrificed taking into account the appearance of necrosis in tumors, weight loss greater than 20% of initial body weight, and tumor volume greater than 2000 mm^3^. Statistical analyses of tumor volumes were performed using two-way ANOVA followed by Sidak’s multiple comparisons, and by log-rank test for survivals.

### 2.7. Toxicity Study

Hematological toxicity was assessed by numeration of red blood cells and platelets on an automated hematology analyzer (Nihon Kohden France, Le Plessis-Robinson, France). Statistical analysis was performed with two-way ANOVA followed by Tukey’s multiple comparison test. Bone marrow, liver, and kidney toxicity was assessed on plasma isolated by centrifugation (10 min at 600× *g*). Each sample was assessed in duplicate. Flt3-ligand concentration was quantified by ELISA (Bio-Techne, Noyal Châtillon sur Seiche, France) following manufacturer’s protocol. ASAT (Bioo Scientific, Austin, TX, USA), ALT, (Bioo Scientific, Austin, TX, USA), urea (BioAssay Systems, Hayward, CA, USA), and creatine (BioAssay Systems, Hayward, CA, USA) were quantified using quantitative colorimetric assays following manufacturer’s instructions. Statistical analysis was performed with two-way ANOVA followed by Sidak’s multiple comparison test.

### 2.8. Statistical Analysis

Statistical analysis were performed using Prism (GraphPad Software Inc., San Diego, CA, USA). A *p-*value of 0.05 or less was considered significant.

## 3. Results

### 3.1. PD-L1 Expression on M113^PD-L1+^ and M113^WT^ Xenograft Tumors

M113^PD-L1+^ melanoma cells were generated by transfection of M113^WT^ parental cells derived from a melanoma patient with *PCDC1* cDNA. PD-L1 expression on both M113^WT^ and M113^PD-L1+^ cells was checked on in vitro cultures by flow cytometry analysis and showed that 99% of M113^PD-L1+^ cells express heterogeneously PD-L1, with 75% expressing high levels and 25% expressing low levels of PD-L1 (Appendix A). M113^WT^ cells were negative. After subcutaneous engraftment in NSG mice flank, M113^PD-L1+^ and M113^WT^ melanoma tumors reached a volume around 80 mm^3^ within 7 days. Such tumor volume appeared suitable to investigate TAT efficacy. PD-L1 expression was confirmed on M113^WT^ and M113^PD-L1+^ melanoma tumors ex vivo by immunostaining and in vivo by immuno-PET (Figure 1). Hematoxylin and eosin staining demonstrated that cell structure was similar in both type of tumors (Figure 1A,B). Immunochemistry staining showed that only M113^PD-L1+^ tumors expressed PD‑L1 (Figure 1E). PD-L1 expression was not recovered in M113^WT^ cells after in vivo implantation (Figure 1F). No stainings were observed with the isotype control (Figure 1C,D). PD-L1 expression was confirmed in vivo by immuno-PET imaging using ^64^Cu-radiolabeled anti-PD-L1 mAb, 1 and 2 weeks (Figure 1G,H), respectively, after tumor implantation. M113^WT^ tumors remained negative (Figure 1I,J). In parallel to PD-L1 expression, we also insured that M113^PD‑L1+^and M113^WT^ xenograft tumors were indeed proliferating at the time of TAT treatment. Therefore, a Ki67 immunohistochemistry staining was performed on those tumors, collected 7 days after implantation in NSG mice. The Ki67 staining demonstrated that cells in both types of tumors were in proliferation (>60%) at the time we planned on initiating TAT (Figure 2 and Appendix A). Staining also showed a central necrotic zone in the M113^WT^ tumor that may have developed early in this xenograft melanoma model (Figure 2B). These results demonstrate that M113^PD‑L1+^ melanoma tumors expressed PD-L1 in vivo and confirmed that PD-L1 was a relevant tumor target to assess TAT efficacy, while M113^WT^ tumors that did not express PD‑L1 represented a suitable control to evaluate targeting specificity. In addition, both tumors were proliferating 7 days after engraftment when TAT was to be performed.

### 3.2. Dose Escalation Study of ^213^Bi-anti-hPD-L1 mAb

The radiochemical yields of ^213^Bi-anti-hPD-L1 mAb were 98.7 ± 1.1% in all experiments, and affinity after radiolabeling was slightly decreased compared to unmodified anti-hPD-L1 mAb (6.5 *×* 10*^−^*^9^ vs. 2 *×* 10*^−^*^9^ M, respectively) but remained in the nanomolar range (Appendix A), which was high and suitable for the study. A dose escalation study was then performed on naïve NSG mice to define the best activities for TAT. Groups of three mice received activities of ^213^Bi-anti-hPD-L1 mAb ranging from 125 to 395 kBq/g. Kaplan–Meier survival curves showed that activities of 395, 335, and 205 kBq/g were highly toxic, with median survivals of 9 and 10 days, respectively (Figure 3A). In these groups, mice were sacrificed based on weight loss that was extremely rapid and greater than 20% of initial body weight (Figure 3B). Instead, groups receiving 125 and 165 kBq/g ^213^Bi-anti-hPD-L1 mAb survived until the end of the study (Figure 3A). Hematologic and biochemistry parameters were determined at the end point for each mouse and compared to status before injection of the radiopharmaceutical (T0). Hematologic toxicity was assessed by platelet and erythrocyte (RBC) counts. At the end point, all the mice injected with 165 to 395 kBq/g of radiolabeled anti-hPD-L1 mAb exhibited a significant and dose-dependent drop of the platelets compared to T0 (Figure 3C). The group that received 125 kBq/g of ^213^Bi-anti-hPD‑L1 mAb demonstrated a slight but not significant decrease in platelet count. No significant change was observed in any group for RBC count (Figure 3D). These results indicated that, except with 125 kBq/g of ^213^Bi-anti-hPD-L1 mAb, all the other injected activities induced significant thrombocytopenia in the animals. In addition to hematologic parameters, bone marrow toxicity was determined by dosing plasma Flt3-ligand concentration (Figure 3E). At T0, median Flt3-ligand concentration in plasma was 190 pg/mL. A considerable and very significant increase in Flt3-ligand concentration was observed in mice injected with 335 and 395 kBq/g of radiopharmaceuticals, with median concentrations reaching 2058 and 2091 pg/mL, respectively. Plasma Flt3-ligand was also increased in the 205 kBq/g group—in particular, up to 1618 pg/mL in one mouse that developed acute toxicity. These results demonstrate that acute toxicity observed in the mice injected with activities ranging from 205 to 395 kBq/g of ^213^Bi-anti-hPD-L1 mAb was associated with bone marrow impairment. Finally, we also observed some increase in Flt3‑ligand concentration in the groups that received 125 et 165 kBq/g of radiolabeled mAb. AST, ALT, and urea were also assessed as biochemical parameters of liver and kidney failure. No change was observed for AST or ALT in any group (data not shown). Urea, instead, was increased in all the groups injected the radiopharmaceutical at activities of 205 kBq/g and greater (Figure 3F). No kidney toxicity was observed in the groups that received 125 and 165 kBq/g of radiolabeled mAb. These results suggested that despite a slight hematologic toxicity on platelets and a minor impact on bone marrow, TAT experiments could eventually be investigated with 125 and 165 kBq/g of ^213^Bi‑anti‑hPD‑L1 mAb.

### 3.3. Assessment of TAT Efficacy Using ^213^Bi-anti-hPD-L1 mAb in M113^PD-L1+^ Melanoma Xenograft Model

We first confirmed that anti-hPD-L1 mAb without modification or radiolabeling had no impact on tumor growth or survival in our preclinical melanoma model by treating M113^PD-L1+^ melanoma tumor bearing mice with 20 and 100 μg of mAb, which were, respectively, 3- to 16-fold the amount of mAb used in TAT experiments (Appendix A). Then, 7 days after subcutaneous graft of M113^PD-L1+^ melanoma cells, when tumors reached an average volume of 80 mm^3^, TAT was performed by i.v. injection of 125 or 165 kBq/g of ^213^Bi-anti-hPD-L1 mAb. Control groups included treatment with 125 or 165 kBq/g of IgG2bκ isotype control radiolabeled with bismuth-213 or injection of 100 μL PBS. TAT efficacy was determined based on tumor growth and survival. Mice treated with 125 and 165 kBq/g of ^213^Bi-anti-hPD-L1 mAb demonstrated a significant and similar tumor growth delay compared to mice treated with the radiolabeled isotype control at 125 or 165 kBq/g or to the PBS control mice (Figure 4A). No difference was observed between these 3 control groups with respect to tumor growth.

More precisely, a few days after TAT and throughout the follow up, M113^PD-L1+^ melanoma development was significantly delayed in mice injected with 125 and 165 kBq/g of ^213^Bi-anti-hPD-L1 mAb compared to PBS control group (**** *p* < 0.0001 for both groups). This impact on tumor progression resulted in improved survival (Figure 4B). Median survival was indeed 64 and 67 days in the groups treated with TAT at 125 and 165 kBq/g of ^213^Bi-anti-hPD-L1 mAb, respectively, compared to PBS control group (MS = 47.5 days, **** *p* < 0.0001) or compared to each relevant ^213^Bi-IgG2bκ isotype control group at 125 kBq/g (MS = 49 days, **** *p* < 0.0001) or 165 kBq/g (MS = 53 days, *** *p* = 0.0008). No significant survival difference was observed between the two groups treated with TAT using radiolabeled anti-hPD-L1 mAb. In addition, weight follow up, as assessed by percentage weight variation compared to initial body weight, did not demonstrate any major variation between the groups treated with radiolabeled mAb or receiving only PBS (Figure 4C). However, one mouse died early after injection of 165 kBq/g of ^213^Bi-anti-hPD-L1 mAb due to undefined reasons, which implies that this activity may induce acute toxicity. All these results demonstrate that TAT using bismuth-213 and targeting PD-L1 was efficient in this melanoma preclinical model and suggest that optimal treatment activity was 125 kBq/g.

### 3.4. Toxicity after TAT Using ^213^Bi-anti-hPD-L1 mAb in M113^PD-L1+^ Melanoma Xenograft Model

Hematologic toxicity was assessed in animals treated with TAT at 125 and 165 kg/g of ^213^Bi-anti-hPD-L1 mAb and in PBS control animals. Platelets and RBC counts were evaluated before tumor engraftment (T0), 20 to 28 days after TAT (intermediate), and at the end point. As already observed during the dose escalation study, we noted a significant decrease in platelet numbers in both groups treated with TAT 20 to 28 days after injection of ^213^Bi-anti-hPD-L1 mAb (**** *p* < 0.0001), and the amount of platelets was not restored at the end point (** *p* = 0.0023, **** *p* < 0.0001) (Figure 5A).

TAT had no impact on RBC count (Figure 5B). Toxicity to bone marrow, liver, and kidneys was also investigated in the mice treated with TAT or receiving only PBS at the end point to compare with status before xenograft (T0). Dosing of Flt3-Ligand in plasma showed a significant increase of its concentration at the end point in animals treated with 165 kBq/g (**** *p* < 0.0001) of ^213^Bi-anti-hPD-L1 mAb (Figure 5C). We observed a similar toxicity with 165 kBq/g of ^213^Bi-isotype control mAb (data not shown). These data are consistent with the dose escalation study results and confirmed that TAT targeting PD-L1 induced some bone marrow toxicity, especially at 165 kg/g, and reinforced the use of 125 kBq/g for therapy. No increase was observed for AST, ALT, or urea after TAT, attesting that this treatment and the activities selected did not impair liver or kidney function (Figure 5D–F).

### 3.5. Assessment of ^213^Bi-anti-hPD-L1 mAb Efficacy in M113^WT^ Melanoma Xenograft Model

Based on anti-tumor response and toxicity study, the previous results demonstrated that TAT should be performed with 125 kBq/g. To confirm that the TAT efficacy we observed was a result of a specific tumor targeting, we repeated the same experiment in mice engrafted with M113^WT^ melanoma tumors that did not express PD-L1. Seven days after M113^WT^ tumor graft, animals were treated with either 125 kBq/g of ^213^Bi-anti-hPD-L1 mAb or ^213^Bi-IgG2bκ isotype control, and control animals received 100 μL of PBS. In this experiment, tumor growth and survival appeared identical in all the groups independent of the treatment received by the animals (Figure 6). These data demonstrate that TAT efficacy in the M113^PD-L1+^ melanoma xenograft model was indeed achieved because of PD-L1-specific tumor targeting.

## 4. Discussion

Melanoma has a high metastatic potential, which makes it the most aggressive and lethal cutaneous cancer. Over the past decade, the development of ICI therapies with blocking antibodies directed against anti-CTLA-4, anti-PD-1, and anti-PD-L1 has totally changed the fate of metastatic melanoma patients thanks to their impressive therapeutic efficacy. However, the objective response rate remains limited to 40%, which means that other therapeutic strategies are still needed [40,41]. TAT is one of the promising treatment strategies currently being developed in oncology [44]. TAT combines the toxicity of an alpha-particle emitter and the specificity of a vector that can be immunologic. Comparison of TAT and TRT with beta particle emitters labelling the same vector, in preclinical and clinical studies, has demonstrated the superiority of alpha-particle emitters in terms of efficacy [45]. These results are related to the physical characteristics of alpha-particles (high LET, short linear path limiting the toxicity to the surrounding healthy tissues) and the fact that their efficacy is not affected by hypoxia or cell cycle status [33].

In this study, we developed and investigated the efficacy of TAT targeting PD-L1 antigen in a preclinical melanoma xenograft model. PD-L1 appeared indeed a very interesting target, as already mentioned, because of its expression on the cell surface of the tumor cells and within the tumor microenvironment, increasing the amount of antigen to target but also because several antibodies are already available for clinical application [19,20,21]. For such purpose, we used a human melanoma cell line expressing stable cell surface PD-L1, M113^PD-L1+^, after transfection of a parental cell line derived from a melanoma patient metastasis, M113^WT^ [46]. This transfection was indispensable since melanoma cells lose expression of PD-L1 after in vitro culture without IFN**γ**—that is, the main inducer of its expression on tumor cells [13]. PD-L1 expression on M113^PD-L1+^ cells and tumors appeared heterogeneous. Such heterogeneity is commonly observed between melanoma patients but also between the different tumor sites within a patient [47]. Despite this heterogeneity, TAT was efficient, which implies that any radiolabeled anti-hPDL1 mAb bound to one cell expressing PD-L1 would be able to destroy the few surrounding tumor cells, and that targeting is feasible regardless of the total amount of antigenic sites in the tumor. In B lymphoma, TRT using radiolabeled anti-CD20 mAbs (Bexxar^®^ and Zevalin^®^) have demonstrated high clinical efficacy in relapsed or refractory patients, and Zevalin^®^ appeared significantly more efficient than immunotherapy with rituximab, with an overall survival rate of 80 vs. 56% (*p* = 0.002) and a complete response rate of 30 vs. 16% (*p* = 0.04) [48]. Because of TRT’s unique mechanisms of action, clinical use of a radiolabeled anti-PD-L1 mAb could be of great interest to overcome the limitations observed with current ICI therapies. Considering the short half-life of alpha emitters, such as bismuth-213 (45 min) or astatine-211 (7 h), anti-PD-L1 TAT could also be used in combination with nivolumab to improve anti-tumor response.

It has been recently shown in vitro and in vivo that PD-L1 expression on a tumor is transiently increased after irradiation and, in particular, when DNA-double strand breaks are induced [49,50,51]. Therefore, it would be interesting to confirm by molecular imaging or other means if TAT with ^213^Bi-anti-hPD-L1 mAb can induce upregulation of PD-L1 expression and then investigate fractionated TAT to further improve treatment efficacy.

One of the major limitations of our xenograft model is the animal immunodeficiency, which is preventing, on the one hand, analysis of TAT impact on the immune cells that constitutively express PD-L1, and, on the other hand, analysis of endogenous immune response. Concerning TAT toxicity, the study by Josefsson et al. is providing interesting dosimetric data. They demonstrated that the spleen was expected to receive the highest activity deposit, followed by the tumor, liver, and thymus and that the dose-limiting organ would be the bone marrow [31]. This supports the feasibility of targeting PD-L1 with TAT since the spleen is not considered as a vital organ, the liver is quite a radioresistant organ, and thymus function is reduced in adults. Concerning the impact on endogenous immune response, it would be interesting to use an immunocompetent melanoma tumor model. Alternatively, since combination therapies may provide synergistic effects and better tumor control, we are currently investigating the potential of combining TAT with ^213^Bi-anti-hPD-L1 mAb and adoptive transfer of tumor-specific T-cells in the same preclinical human melanoma model.

## 5. Conclusions

This study showed that TAT targeting PD-L1 in a human melanoma xenograft model was associated with efficient anti-tumor response, as demonstrated by significant delay in tumor growth and improved survival with minimal hematologic toxicity. This demonstrates that anti-PD-L1 antibodies could be used as theranostics in molecular imaging to select patients for ICI therapy and assess response to treatment but also in TAT to target the tumor and its stroma.

## Figures and Tables

**Figure 1 cancers-13-01256-f001:**
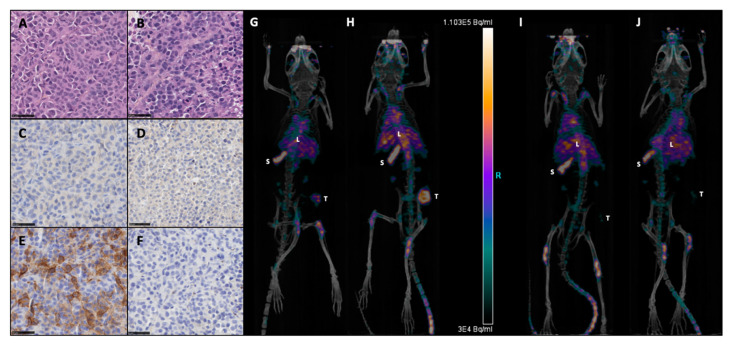
Expression of programmed death-ligand 1, B7-H1, CD274 (PD-L1) on M113 melanoma xenograft tumors. Formalin-fixed and paraffin-embedded histologic sections of M113^PD-L1+^ (**A**,**C**,**E**) and M113^WT^ (**B**,**D**,**F**) melanoma xenograft tumors were examined immunohistochemically after hematoxylin and eosin (**A**,**B**), rabbit isotype control (**C**,**D**), and anti-hPD-L1 (**E**,**F**) staining. All sections were photographed at 40x original magnification, scale bars = 50 µm. Data are representative of 4 different M113^PD-L1+^ and M113^WT^ tumors. Immuno-PET imaging of mice bearing subcutaneous M113^PD-L1+^ (**G**,**H**) or M113^WT^ (**I**,**J**) tumors at 1 (**G**,**I**) or 2 weeks (**H**,**J**) after tumor engraftment and 48 h after injection of 10 MBq ^64^Cu-anti-hPD-L1mAb (L: liver, S: spleen, T: tumor). Data are representative of 3 different mice in each group.

**Figure 2 cancers-13-01256-f002:**
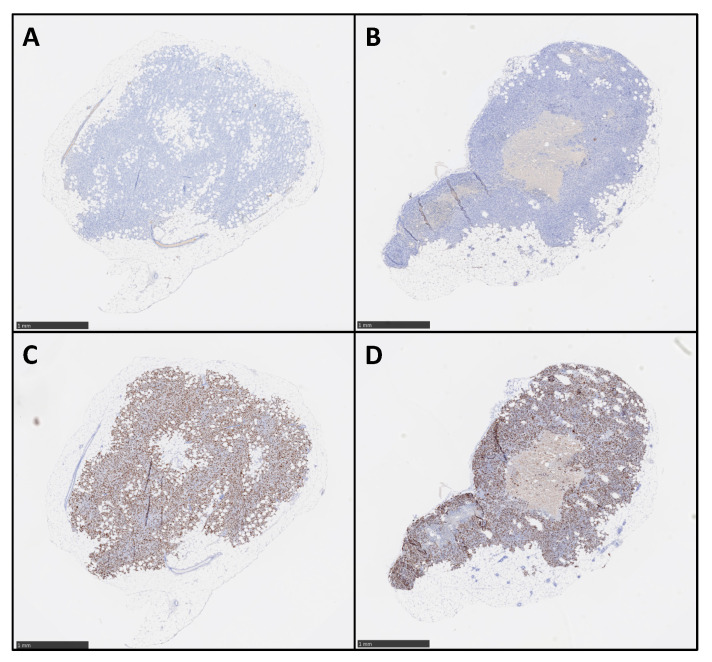
Ex vivo proliferation analysis of M113 melanoma xenograft tumors**.** M113^PD-L1+^ (**A**,**C**) and M113^WT^ (**B**,**D**) melanoma tumor proliferation was examined on formalin-fixed and paraffin-embedded histologic sections after immunohistochemical staining with mouse anti-human Ki67 mAb (**C**,**D**) or mouse IgG1κ isotype control (**A**,**B**). Data are representative of 4 different M113^PD-L1+^ and M113^WT^ tumors. Scale bars = 1 mm.

**Figure 3 cancers-13-01256-f003:**
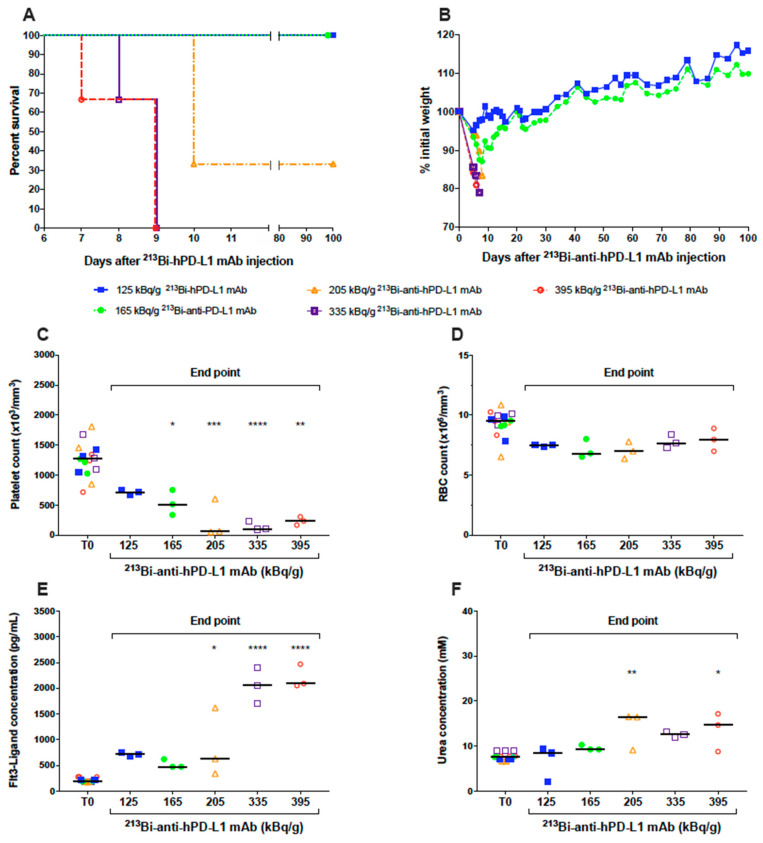
Dose-escalation study of ^213^Bi-anti-hPD-L1. Naive NSG mice (*n* = 3 per group) received an i.v. injection of 125 (
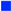
), 165 (
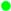
), 205 (
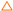
), 335 (
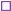
), 395 (
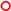
) kBq/g ^213^Bi-anti-hPD-L1 mAb. (**A**) Kaplan–Meier survival analysis. (**B**) Mean weight variation in each group, as expressed as percent of initial weight. Activities ranging from 205 to 395 kBq/g ^213^Bi-anti-hPD-L1 mAb induced acute toxicity as demonstrated by weight loss > 20% of initial weight and resulting in mouse sacrifice. Animals surviving acute toxicity were followed for a 100-day period before euthanasia. (**C**) Platelet, (**D**) RBC counts, as well as (**E**) plasma FLT3-Ligand and (**F**) urea concentrations were monitored for each animal before injection of radiolabeled anti-hPD-L1 mAb (T0) and at end point. Each sample was assessed in duplicate. Bar represents the median. Activities ranging from 205 to 395 kBq/g ^213^Bi-anti-hPD-L1 mAb induced significant toxicity on platelets (* *p* = 0.0297, ** *p* = 0.0023, *** *p* = 0.0001, **** *p* < 0.0001), bone marrow (* *p* = 0.0163, **** *p* < 0.0001), and kidneys (* *p* = 0.0297, ** *p* = 0.0045). Statistical analysis was performed with two-way ANOVA followed by Sidak’s multiple comparison test.

**Figure 4 cancers-13-01256-f004:**
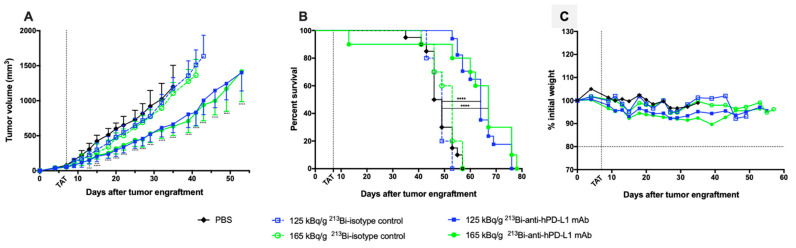
Targeted alpha-particle therapy (TAT) efficacy using ^213^Bi-anti-hPD-L1 mAb in M113^PD-L1+^ melanoma xenograft model. At day 0, NSG mice were engrafted subcutaneously with 1×10^6^ M113^PD-L1+^ melanoma cells. At day 7, TAT was performed by i.v. administration of 125 kBq/g ^213^Bi-Anti-hPD-L1 mAb (
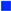
, *n* = 17), 165 kBq/g ^213^Bi-Anti-hPD-L1 mAb (
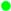
, *n* = 10), 125 kBq/g ^213^Bi IgG2b isotype control (
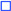
, *n* = 10), 165 kBq/g ^213^Bi IgG2b isotype control (
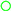
, *n* = 10), or PBS for control animals (**◆**, *n* = 20). (**A**) Tumor volume, represented by mean and SD, was determined sequentially from engraftment until signs of tumor necrosis or volume reached 2000 mm^3^ and animals were sacrificed. Compared to PBS and isotype control groups, TAT significantly delayed tumor growth in mice treated with 125 kBq/g ^213^Bi-Anti-hPD-L1 mAb or 165 kBq/g ^213^Bi-Anti-hPD-L1 mAb (* *p* = 0.0313, *** *p* = 0.0007, **** *p* < 0.0001). Statistical analysis was performed with two-way ANOVA followed by Sidak’s multiple comparison test. (**B**) Kaplan–Meier survival analysis. TAT with 125 kBq/g and 165 kBq/g ^213^Bi-Anti-hPD-L1 mAb significantly increased survival (MS = 64 and 67 days, respectively) compared to PBS control group (MS = 47.5 days, **** *p* < 0.0001). Survival was not significantly different in both isotype control groups compared to PBS control group. The *p*-values were determined by log-rank test. (**C**) Mean weight variation in each group is expressed as percent of initial weight.

**Figure 5 cancers-13-01256-f005:**
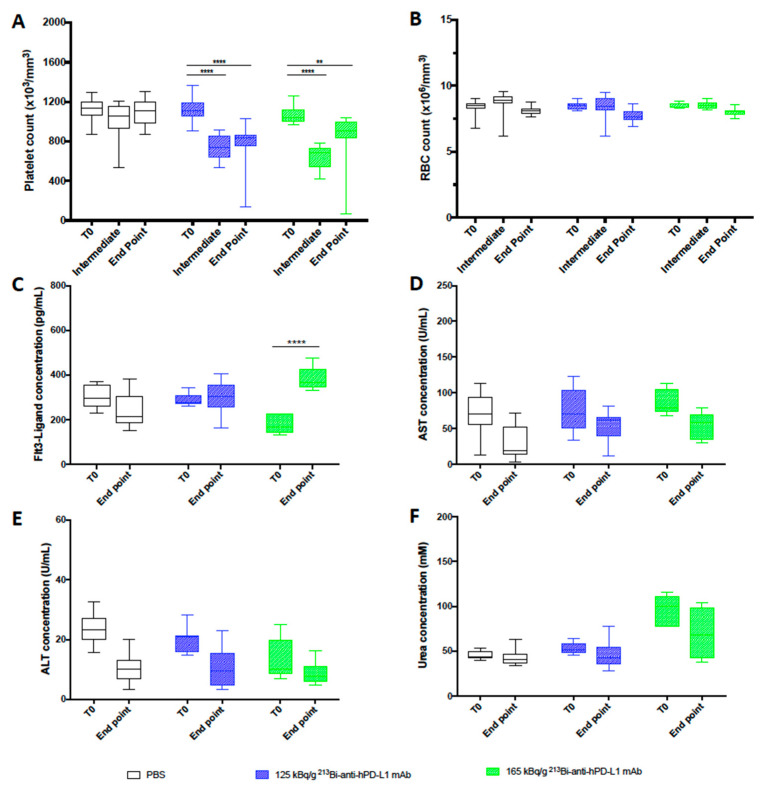
Toxicity study after TAT using ^213^Bi-anti-hPD-L1 mAb mAb in M113^PD-L1+^ melanoma xenograft model. (**A**) Platelet and (**B**) RBC counts, as well as plasma (**C**) FLT3-Ligand, (**D**) AST, (**E**) ALT, and (**F**) urea concentrations were assessed in animals receiving TAT treatment with 125 kBq/g ^213^Bi-Anti-hPD-L1 mAb (
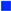
), 165 kBq/g ^213^Bi-Anti-hPD-L1 mAb (
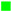
) or receiving PBS (☐). CBC were performed before TAT (T0), 20 to 28 days after TAT (Intermediate), and at end point. Other toxicity parameters were assessed in duplicate before TAT (T0) and at end point. Box extends from the 25th to 75th percentiles, line represents the median and the whiskers go down to the smallest value and up to the largest. Statistical analysis was performed with two-way ANOVA followed by Tukey’s multiple comparison test (** *p* = 0.0023, **** *p* < 0.0001).

**Figure 6 cancers-13-01256-f006:**
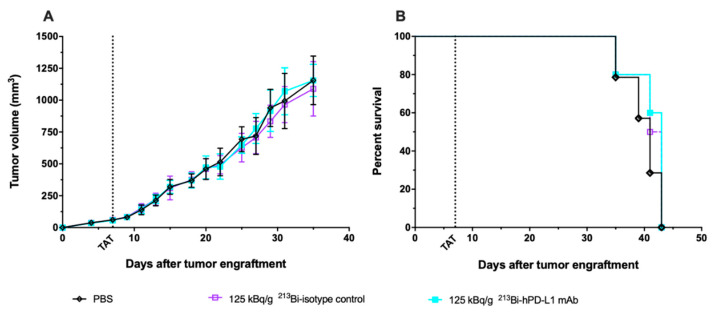
TAT using ^213^Bi-anti-hPD-L1 mAb in PD-L1 negative M113^WT^ melanoma xenograft model. At day 0, NSG mice were grafted subcutaneously with 1×10^6^ M113^WT^ melanoma cells that did not express PD-L1. At day 7, TAT was performed by i.v. administration of 125 kBq/g ^213^Bi-Anti-hPD-L1 mAb (
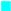
, *n* = 10), 125 kBq/g ^213^Bi IgG2b isotype control (
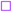
, *n* = 10), or PBS for control animals (◇, *n* = 14). (**A**) Tumor volume, represented by mean and SD, was determined sequentially from engraftment until signs of tumor necrosis or volume reached 2000 mm^3^ and animals were sacrificed. No difference was observed in tumor growth between the different groups. (**B**) Kaplan–Meier survival analysis. Treatment with 125 kBq/g ^213^Bi-Anti-hPD-L1 mAb or ^213^Bi IgG2b isotype control had no impact on survival (MS = 43 and 42 days, respectively) compared to PBS control group (MS = 41 days).

## Data Availability

The data presented in this study are available on request from the corresponding author.

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
