# Peer review of "Anti-Tumor Efficacy of PD-L1 Targeted Alpha-Particle Therapy in a Human Melanoma Xenograft Model"

_cancers, 2021, doi:10.3390/cancers13061256_

Round 1

Reviewer 1 Report

The manuscript entitled "Anti-tumor efficacy of PD-L1 Targeted-alpha-therapy in a 2 human melanoma xenograft model" addresses the interesting issue of alpha rays emitting isotopes linked to a antibody specific for tumor antigens, in this case PD-L1. 

Although highly interesting the paper has significant flaws. 

The only toxicity that author take into account is a general toxicity relative to TAT targeting of bismuth-213 at different dosages, not that of bismuth-213 plus anti-hPDL1. In the immune deficient mice model used, the expression of hPDL1 is restricted to t human melanoma cell lines which are transduced to express it. The murine cells of  NSG mice do not express human PDL1 and would therefore not be targeted by the TAT/bismuth-213. So the only effect observed is that of direct targeting of tumor cells by bismuth-213. No information on toxicity on normal cells is availbale yet to be expected since PDL-1 expression in men is not restricted to tumor cells but is expressed on a variety of normal and immune cell types. Tumor cells have adopted this PD-1/PD-L1 mechanism to suppress immune surveillance and facilitate growth. Authors must therefore demonstrate the efficacy of the proposed therapeutic intervention and the absence of toxic side effects in a syngeneic model using murine PDL1 and anti mPDL1 radiolabeled. This is crucial for the evalutation of the TAT approach and must therefore be presented in this paper. 

Given the specificity of the intervention used it is surprising that the therapy only delayed tumor growth. It would be interesting to analyze ex vivo cells recovered from the tumors for PDL1 expression.

Minor comments:

The figure legends should be inserted under the figures.

There are several typos in the manuscript, for example:

“..if PD-L1 could be also be a good target for treatment using targeted..” line 19 page 1

“..is correlated with a poor prognostic prognosis in melanoma. Anti-PD-L1 mAbs have been..” line 25 page 1

“..point in animals treated with 165 kg/g (****p<0.0001) of 213Bi-anti-hPD-L1 mAb ….” lane 321

Reviewer 2 Report

The article “Anti-tumor efficacy of PD-L1 Targeted-alpha-therapy in a human melanoma xenograft model ” shows that targeted alpha-therapy with PD-L1 as target reduces tumor growth in an immunosuppressive melanoma mouse model. The manuscript is well written and provides clearly represented figures.

Minor comments:

  • Please include in the introduction the aspect that the conclusions from multiple trials with PD-L1 as biomarker are not consistent and that the predictive biomarker role of PD-L1 has limitations (line 64-65).
  • The authors might want to indicate the percentage of low and high PD-L1 expression in the Figure S1.
  • In Figure S2 it seems that for the grey curve some SD are missing.
  • Please comment why in Figure 6 tumors where only kept until a tumor growth of around 1000 mm3 and not as state in the figure legend and as done for all the other experiments until 2000 mm(line 394).
  • The experiments show high specificity for the antibody fusion and is thus a nice proof of concept. Please comment on the potential clinical additional value of the antibody fusion. What is the benefit for patients to use the fusion antibody in comparison to nivolumab treatment?
  • This is only a comment, no action needed: In the discussion, the authors state that using immunocompetent mice would have been even better. For future experiments the authors might also think about if starting treatment at a tumor size around 80 mm3 really represents the clinical setting, or if they should start treatment later.
  • Line 19: Please remove “be” in front of “also”.
  • Line 155: Please use singular for “scanner”.
  • Line 294: Please specify which p-value belongs to what.
  • Line 354 / Fig. 2: Please specify which panels are Ki67 mAb and which are isotype control.
  • Line 368 / Fig. 4: Please correct grammatically the first sentence.
  • Line 417: Full stop is missing.

Reviewer 3 Report

This well written manuscipt explores in depth and with necessary controls an interesting approach to circumvent non-response to checkpoint therapy by using PD-L1 as ‚vehicle‘ to transport the tumor-toxic substance instead of continous antibody applications as therapy.

One major caveat is the layout: Generally figure legends are next to the figure and not in a separate section. This would ease reading substancially.  

This reviewer has two questions regarding the characterization of the tumors with/without PDL1 expression: 1. Hast he Ki67 expression been quantified? 2. Is central necrosis is occuring in PDL1+ tumors, too? But this is not really visible in the image shown (due to image reproduction quality).  

In addition it would be helpfull to indicate the area that has been enlarge to show K67 staining.

While the authors state that PDL1 expression may be lost during cell culture, and therefore a transfected clone is needed, hast he stability of PDL1 expression over 40-80days (time of tumor engraftment) been tested (e. g. in vitro cultures without G418)?

For radiotoxicity the authors analyse RBC and platelet counts. In FIgure 3 platelet count is C and RBS is D, whle in the legens it is vice versa. Please correct. In addion, in Figure 3D there seems to be an effect in all treatments. Can thrombocytes serve as sole marker for leucopenia (or should this read thrombocytopenia instead)?

What was the amount of radiolabelled antibody administered? 5µg/mouse? (see lines 281/282)

As the authors state, their data lead to  two aspects to be further investigated: a syngenic and therefore immunocompetent model and a regimen using multiple radiation doses. Nevertheless this manuscript comprises an complete set of experiments with important results.

Minor points:

Line 48: a comma is missing (after ‚melanoma‘)

Line 68: clinical trial should be in plural

Line 338: This should bie Fgure 6 (there is no Figure 7).

Figure S2: The authors should explain, how they could calculate a SD with an n=2

Reviewer 4 Report

Dear Editor, it was a pleasure to review the manuscript by dr Capitao et al.

The paper analyses the efficacy and toxicity of 213Bi-anti-human-PD-L1 mAb in M113 melanoma cell line transfected with PDL1, implanted in NSG mice.

Firstly, the authors use only 1 cell line, M113, from which they derived the PDL1 overexpressed M113PDL1+. At least another cell line overexpressing PDL1 should have been used to confirm the reproducibility of the results. 

Moreover, the authors analyze the role of the 213Bi-anti-human-PD-L1 mAb in NSG mice: when evaluating the toxicity of this treatment, the disruption of Tcell activity, and its consequences when targeting PDL1, may not be correctly assessed. Anemia and thrombocytopenia were included in the toxicity profile, leucopenia should be expected, and the importance of such toxicity may be crucial. 

Finally, the activity of the anti-human-PD-L1 mAb should be confronted to the anti-human-PD-L1 mAb in a context of immuno-competency, to assess if the higher toxicity related to the TAT-mAb increases the responses already obtained with anti-PD-L1 alone.

Round 2

Reviewer 1 Report

The authors have adequately addressed all issues raised. The various modification made to the manuscript respond to the toxicity issue. Therefore I can recommend publication. 

Author Response

No further response requested

Reviewer 4 Report

The authors did not satisfy the reviewer's requests.

Author Response

No further response requested